# Natural Born Killers: NK Cells in Cancer Therapy

**DOI:** 10.3390/cancers12082131

**Published:** 2020-07-31

**Authors:** S. Elizabeth Franks, Benjamin Wolfson, James W. Hodge

**Affiliations:** Laboratory of Tumor Immunology and Biology, Center for Cancer Research, National Cancer Institute, National Institutes of Health, Bethesda, MD 20892, USA; frankssae@mail.nih.gov (S.E.F.); ben.wolfson@nih.gov (B.W.)

**Keywords:** natural killer cells, NK cells, adoptive cell transfer, NK-92, CAR-NK, haNK, t-haNK

## Abstract

Cellular therapy has emerged as an attractive option for the treatment of cancer, and adoptive transfer of chimeric antigen receptor (CAR) expressing T cells has gained FDA approval in hematologic malignancy. However, limited efficacy was observed using CAR-T therapy in solid tumors. Natural killer (NK) cells are crucial for tumor surveillance and exhibit potent killing capacity of aberrant cells in an antigen-independent manner. Adoptive transfer of unmodified allogeneic or autologous NK cells has shown limited clinical benefit due to factors including low cell number, low cytotoxicity and failure to migrate to tumor sites. To address these problems, immortalized and autologous NK cells have been genetically engineered to express high affinity receptors (CD16), CARs directed against surface proteins (PD-L1, CD19, Her2, etc.) and endogenous cytokines (IL-2 and IL-15) that are crucial for NK cell survival and cytotoxicity, with positive outcomes reported by several groups both preclinically and clinically. With a multitude of NK cell-based therapies currently in clinic trials, it is likely they will play a crucial role in next-generation cell therapy-based treatment. In this review, we will highlight the recent advances and limitations of allogeneic, autologous and genetically enhanced NK cells used in adoptive cell therapy.

## 1. Introduction

Harnessing the immune system for cancer treatment is one of the most exciting therapeutic possibilities in the history of cancer treatment, and one of the oldest. While the idea of activating the immune system through outside agents to boost an anti-cancer response was first applied in the late 1800s, it has taken a century for similar findings to be validated and applied in contemporary medicine. Modern immunotherapy comprises two broad strategies: treating patients with therapeutic agents that are capable of engaging, expanding and enabling autologous immune cells within the body, and directly modifying effector immune cells to promote their cytolytic ability. Direct cellular modification can take place within the patient through in vivo gene therapy techniques (although no treatment using these methods has gained FDA approval), or by isolating the target cell population and manipulating it ex vivo. 

Ex vivo manipulation of immune cells has gained renown through the development and application of chimeric antigen-receptor T cells (CAR-T). Through this process, T cells harvested from a healthy donor (allogeneic) or from the patient (autologous) are expanded and genetically modified ex vivo and re-introduced into that patient [1]. Two CAR-T therapies have been granted FDA approval as of this writing: tisagenlecleucel in relapsed/refractory B cell precursor acute lymphoblastic leukemia (ALL) [2], and axicabtagene ciloleucel in relapsed/refractory diffuse large B cell lymphoma [3], both specific for the B cell antigen CD19. While these treatments have gained success and there are a multitude of ongoing clinical trials using CAR-T, rates of long-term progression-free survival for CAR-T patients are low, frequently attributed to low CAR-T cell persistence in vivo and tumor-associated antigen (TAA) modulation or loss (reviewed here: [4]). One strategy to circumvent antigen escape is through the use of a cytolytic cell that functions independently of antigen: the natural killer (NK) cell. 

## 2. Natural Killer Cells

Natural killer cells are large granular lymphocytes that make up approximately 10–15% of the peripheral blood lymphocyte population, provide a rapid response to viral infection and participate in anti-tumor immune surveillance. In contrast to T cells, NK cell anti-tumor activity does not require antigen recognition in complex with MHC; instead, it is activated following a lack of recognition of “self” markers on tumor cells coupled with a combination of competing activating and inhibitory receptors. There are three major classes of NK receptors: killer immunoglobulin-like receptors (KIRs), the primary MHC-I receptors, C-type lectin receptors that recognize non-classical MHC-I or MHC-I-like molecules, and the natural cytotoxicity receptors. Through the process of NK cell education, the expression of different activating or inhibitory receptors is tuned to prevent NK cells from targeting the body’s own cells while promoting their recognition of non-self cells. Several models of how NK education occurs include licensing/arming, disarming, rheostat and tuning. While the full mechanisms of NK cell education are beyond this article, they have been well reviewed by other authors [5]. Mature NK cells are capable of rapidly recognizing the difference between self and non-self cells. Under healthy conditions, inhibitory NK receptors recognize MHC on the target cells and prevent a cytotoxic response. When a cell is infected with virus or is transformed, downregulation of MHC proteins prevents this inhibitory reaction, and NK cells are activated and subsequently lyse the target cell.

### NK Cell Cytotoxicity

NK cells have two major cytotoxic mechanisms, granulocyte apoptosis mediated by perforin and granzyme and antibody-dependent cell-mediated cytotoxicity (ADCC). Upon NK cell activation, granulocytes are exocytosed, allowing granulocytic perforin to form pores in the cell membrane of target cells [6]. Following cell membrane disruption, granzyme serine proteases are delivered to the cytoplasm of the cell, where they induce apoptosis. 

NK cells also induce ADCC, a mode of cell death in which the cell surface receptors Fc𝛾RIIC (CD32c) and Fc𝛾RIIIA (CD16a) bind the Fc portion of antibodies which have bound target antigen. After engaging the Fc receptor, NK cells can induce cell death through granulocytic apoptosis, engagement of tumor-necrosis factor death receptor signaling (mediated by TRAIL proteins) and release of pro-inflammatory cytokines to help generate an adaptive immune response and promote lysis by additional immune cells [7]. Different antibody isotypes generate various degrees of ADCC, allowing this mechanism to be coopted by certain therapeutic antibodies. For example, if a therapeutic antibody is an IgG1 or IgG3, it can bind the Fc𝛾 receptors on NK cells, and mediate strong ADCC activity [8]. In addition to serving as one mechanism for antibody cytotoxicity, it is also a potential strategy for additive therapeutic benefit between monoclonal antibodies (mAbs) and cell therapies using NK cells. 

## 3. Endogenous NK Cells

The earliest work promoting the NK cell-mediated innate immune response as a tumor therapy focused on the patient’s endogenous NK cell population. These strategies include activating endogenous cells, as well as sensitizing tumor cells to NK-mediated killing. Three general strategies have been developed to promote NK cell-mediated tumor lysis: (1) blocking inhibitory NK cell receptors to increase cytotoxicity against tumors that do not downregulate MHC/HLA proteins, (2) treating patients with NK cell-activating cytokines to promote their expansion, activation and cytotoxicity, and (3) treating patients with additional therapies that result in immunogenic modulation, sensitizing the tumor to NK cell killing. 

### 3.1. Inhibitory Receptor Blocking

Similar to checkpoint blockade antibodies, therapeutic mAbs have been designed to block inhibitory receptors on NK cells. Several inhibitory receptor blocking antibodies are currently in clinical trials; however, it is a novel strategy that has yet to come to fruition. Two blocking antibodies are being investigated that interact with the receptor KIR2D. One, IPH2101, was demonstrated to be effective in vitro and in vivo, where it enhanced NK-mediated killing of KIR matched tumor cell lines and enhanced ADCC [9,10]. However, while it was demonstrated to be safe in a Phase I clinical trial of patients with smoldering multiple myeloma, it was not found to be effective in a follow-up Phase II trial [11]. Further studies are ongoing using IPH2101 in combination with lenalidomide, a multiple myeloma medication that has been shown to expand NK cell populations [9]. Similar to IPH2101, a second KIR2D mAb, IPH2102, was demonstrated to be safe in a Phase I trial of hematologic and solid cancers, but no clinical efficacy was observed as a monotherapy in a Phase II trial [12]. Furthermore, a mAb specific for the inhibitory receptor NKG2A increased NK cell killing of leukemia in vivo [13] and was found safe in a Phase I trial in gynecologic cancers [14].

In addition to NK-specific inhibitory receptors, NK cells also express classical checkpoint receptors, including CTLA-4, PD-1, TIGIT, LAG-3 and TIM-3. Inhibition of these negative signaling regulators using checkpoint blockade antibodies may therefore increase their activation state and cytolytic abilities [15]. Numerous clinical trials are underway investigating the effects of checkpoint inhibitors on NK cells, and have been reviewed elsewhere [15]. 

### 3.2. Cytokines to Activate Endogenous NK cells

Treatment with exogenous cytokines is one of the most frequent immunotherapeutic strategies, with numerous agents being investigated for clinical use in multiple indications and combinations. Interleukins -2, -12, -15, -18 and -21 have been shown to promote NK cytotoxicity and proliferation in vitro and in vivo [16], with IL-2 approved for use in metastatic renal cell carcinoma and metastatic melanoma [17], and approved as a monotherapy in melanoma treatment [18]. Cytokines also play an important role in combination with other immunotherapies and in the activation of allogeneic NK cells. Additionally, several novel immunocytokines are currently in development that promise improved cytokine specificity and activity. The IL-15 superagonist N-803 promotes enhanced NK cell function in vitro and in vivo [19] and has demonstrated safety in early stage clinical trials [20]. Phase I and II trials are currently underway in multiple cancer indications in combination with numerous immuno-oncology agents. Furthermore, NHS-IL-12 is a fusion of IL-12 molecules to a tumor necrosis-targeting human IgG1 that is currently in Phase I and II combination therapy trials; however, its ability to promote NK cell activity is still under investigation [21].

### 3.3. Immunogenic Modulation

It has been previously reported that certain standard-of-care therapies are capable of changing tumor cell phenotype to sensitize them to killing by cytotoxic immune cells including T cells and NK cells [22,23]. This process has been demonstrated with chemotherapies [24,25,26], endocrine deprivation [27,28], sublethal doses of radiation [29,30,31,32], as well as cytotoxic small molecules, including but not limited to trebananib, ARI-4175, olaparib and bortezomib [33,34,35,36]. These findings present a rationale for combining standard-of-care therapies with a variety of adoptive cell transfer strategies. 

## 4. Adoptive Cell Transfer

NK cells can be obtained from a variety of sources within a patient or healthy donor. While mature NK cells can be isolated from the blood, they can also be differentiated from stem cells, including those isolated from umbilical cord blood, embryonic stem cells and induced pluripotent stem cells. NK cells make up approximately 10–15% of peripheral blood lymphocytes. To obtain clinical grade NK cells, leukapheresis products are first sorted for CD3^−^CD56^+^ cell populations, which are then expanded through cytokine treatment, most commonly IL-2 and IL-15 [37,38]. In addition to expanding the NK cell population, these cytokines also activate NK cells, resulting in increased cytotoxicity [16]. Coculture of isolated NK cells with irradiated feeder cell populations has also proven effective and was demonstrated to be safe in a Phase I clinical trial in advanced digestive cancer, and therefore may be a more efficient method for rapid NK cell expansion [39]. 

NK cells isolated from stem cells benefit from increased ease of storage and purity; however, functional differences from peripheral blood mononuclear cell (PBMC)-derived NK cells have been reported. Notably, in comparison to PBMC-derived NK cells, cord blood NK cells lack expression of the activation marker CD57, have higher expression of the inhibitory receptor NKG2A and express fewer inhibitory KIRs [40], leading to decreased cytotoxicity. However, similar to PBMC isolated NK cells, treatment with IL-2, IL-15, IL-7 or feeder cells has been demonstrated to increase their cytotoxicity [41], indicating that cord blood NK cells remain a viable option [42]. Similarly, a multitude of protocols exist for the differentiation of NK cells from embryonic or pluripotent stem cells that rely on treatment with cytokines or feeder cells [43,44]. Stem cell-derived NK cells may be advantageous over their cord blood counterparts due to their similar cytotoxicity profile in comparison to those isolated from PBMCs [45] and the ability to be maintained as a renewable source of cells [46]. 

In the clinic, adoptive cell transfer of NK cells can use either autologous NK cells (Figure 1A), isolated or generated from the patient’s own blood or stem cells, or allogeneic NK cells (Figure 1B), in which the NK cells are obtained from a healthy donor.

### 4.1. Autologous NK Cells

While in vitro and in vivo studies have widely demonstrated the cytotoxicity of ex vivo expanded NK cell populations against tumor cell lines, clinical results have been less conclusive. In a Phase I/II trial of breast and lung cancer, infusions of ex vivo IL-2‒activated NK cells were compared to injection of a bolus of IL-2. Infusion of activated NK cells and the IL-2 bolus were demonstrated safe, and cytolytic function of freshly isolated PBMCs post-treatment were elevated in a treatment-agnostic manner. However, neither treatment translated to improved patient outcome [47]. A similar clinical trial in malignant glioma infused patients with ex vivo IL-2‒activated NK cells in combination with intravenous injection of low-dose interferon beta. Nine of sixteen (56%) patients exhibited a decrease in tumor growth, with the decrease persisting for over four weeks in 3/16 (19%) patients [48]. In metastatic colorectal cancer and lung cancer, Krause et al. incubated peripheral blood lymphocytes with IL-2 in combination with a heat shock protein 70 (Hsp70) peptide. Hsp70 has been shown to be upregulated on tumor cells, and in vitro incubation of NK cells with Hsp70 has proven to increase the cytotoxicity of NK cells specifically against Hsp70 membrane-positive tumors [49]. The Phase I study reported no toxicities and an increase in ex vivo NK cell cytolytic activity against Hsp70 positive colon carcinoma cells in NK cells from 10/12 patients. Despite this, only 1/12 patient exhibited stable disease during therapy, with another patient showing stable disease in one metastasis and progression in others [49]. A Phase II trial investigating autologous Hsp70-peptide activated NK cells in non-small cell lung cancer is currently underway (NCT02118415). Similarly, patients with metastatic melanoma and renal cell carcinoma were infused with NK cells activated with IL-2. Interestingly, NK cells were found to persist in peripheral circulation for between one week and several months, yet no clinical responses were observed [50]. 

While these studies and others have reported that adoptive transfer of autologous NK cells is safe, they also observed an extremely limited clinical benefit [51]. It is likely that the failure of adoptively transferred autologous NK cells is at least partially due to NK cell recognition of self-HLA molecules on tumor cells, resulting in decreased cytotoxicity. Interestingly, a recently completed clinical trial reported increased efficacy of autologous NK cell transfer in patients with epidermal growth factor receptor (EGFR)-mutated advanced lung adenocarcinoma compared to EGFR wild-type. It was postulated this was due to decreased HLA expression in the EGFR-mutant tumors, shifting the balance of inhibitory and activating receptors acting in concert on the NK cells [52]. Additionally, increased clinical success has been reported when combining NK cells with agents that induce immunogenic modulation such as bortezomib, a reversible proteasome inhibitor [36]. While work with autologous NK cells continues and approximately 50 clinical trials are currently ongoing, investigations into the efficacy of allogeneic NK cell transfer are also underway and have demonstrated significant clinical potential.

### 4.2. Allogeneic NK Cells

Allogeneic NK cell transfer for cancer treatment rose serendipitously from observations made when patients with leukemia were given allogeneic hematopoietic stem cell transplantation (HSCT). NK cells are the first immune subset to recover, and they mediate a patient’s immune response prior to expansion of T cell subsets [53]. T cell populations are the primary mediators of graft versus host disease (GVHD), one of the primary toxicities associated with HSCT. It was hypothesized that these toxicities eclipsed the benefit of NK cell alloreactivity [54], and that allogeneic transplantation designed around donor versus recipient NK cell alloreactivity may decrease acute myeloid leukemia (AML) relapse and protect against GVHD [55]. In a study of 130 patients with hematologic cancers, those who had KIR ligand incompatibility to their donors had increased probability of disease-free survival (87% vs. 39%), transplant-related mortality (6% vs. 40%) and relapse rates (6% vs. 21%) [56]. 

As autologous NK cells suffer from resistance due to recognition of self-HLA signals, selecting HLA-mismatched NK cell donors would remove the inhibitory signal delivered to the allogeneic NK cells, resulting in effective tumor lysis. Infusion of haploidentical NK cells alone was first tested by Miller et al., who demonstrated that injecting immune-suppressed AML patients with NK cells, and subsequent IL-2 treatment, resulted in a rise in IL-15, expansion of the NK cell population, and disease remission in 5/19 patients [57].

These findings have been replicated in additional cohorts of patients with leukemia [58,59,60], lymphoma [61], as well as solid tumors, including a Phase II study in recurrent ovarian and breast cancer [62], and a Phase I study in advanced non-small cell lung cancer [63]. These studies are described in detail in a recent review by Veluchamy et al. [64]. While adoptive cell therapy using allogeneic NK cells is safe and has demonstrated clinical efficacy in the context of hematological malignancies, the results observed in solid tumors are much less impressive [64].

Adoptive cell therapy of allogeneic NK cells suffers from shortcomings similar to that with autologous NK cells, namely the timely ex vivo expansion and activation of clinical grade NK cells and persistence in vivo post-inoculation. While allogeneic cells may be banked and used when necessary, they do not match the ease of using established cell lines [65].

### 4.3. Immortalized NK Cells

Several NK cell lines have been established as potential therapeutic products, including NKG [51], KHYG-1 [66], NK-YS [67], YT [68], YTS [69], NK3.3 [70], NKL [71], and NK-92 [72]. To date, only NK-92 have entered FDA approved clinical trials [16]. NK-92 is an NK-like cell line derived from a 50-year-old male patient with non-Hodgkin’s lymphoma [72], and therefore must be irradiated prior to infusion into recipient patients as to not cause secondary tumorigenesis. In fact, adoptive transfer of irradiated NK-92 cells still shows clinical benefit, with some patients experiencing durable responses. These data further support their use in patients as irradiation increases their safety and reduces off-target effects through diminished survival and inability to proliferate. 

NK-92 cells possess unique characteristics that make them advantageous over naturally occurring NK cells for use as an adoptive cell therapy. They can be readily expanded ex vivo and do not exhibit any variation in subtype or phenotypic characteristics, which is a concern when using allogeneic and autologous NK cells. Furthermore, they lack most KIR receptors but remain armed with a vast array of activating receptors [73]. This increases their cytotoxic potential against the majority of tumor cell lines in vitro and in vivo. NK-92 cells mediate their anti-tumor effect through perforin and granzyme B activity, and have demonstrated efficacy against leukemia, lymphoma, melanoma, prostate and breast cancer [74]. 

In a first-in-human Phase I trial of 12 patients with melanoma or renal cell carcinoma, inoculation of irradiated NK-92 cells proved safe with one minor response [75]. NK-92 cells have since been used in three additional clinical trials in AML [76], hematological malignancies following relapse after autologous HSCT [77], and diverse solid tumors/sarcomas [78]. NK-92 cells have been demonstrated to be safe in all contexts, including when up to 20 repeated infusions were delivered [76]. There are currently three recruiting/active and not recruiting Phase I/II and Phase II clinical trials using NK-92 cells in combination therapies in Merkel cell carcinoma (NCT02465957), pancreatic cancer (NCT03136406), and myelodysplastic syndrome, leukemia, lymphoma or multiple myeloma (NCT02727803). 

NK-92 cells also possess disadvantageous qualities, many of which have been engineered out in later iterations of the cell line. NK-92 cells require exogenous IL-2 for survival and die within 72 hours of cytokine removal [72]. Because of the extreme toxicity and inadvertent stimulation of regulatory T cells (Tregs) associated with systemic administration of IL-2 in human patients, NK-92 cells were engineered to express endogenous endoplasmic reticulum-retained IL-2 [79]. Furthermore, NK-92 cells lack surface expression of CD16, eliminating the ability to mediate ADCC [72]. There are two allelic variants of CD16 created by a phenylalanine to valine amino acid substitution at position 158, with only 10% of the human population carrying the high affinity receptor (158V). To return ADCC competency to the cell line, NK-92 cells have been engineered to express the high affinity CD16/FcγRIIIA (158V) allele (termed high-affinity NK cells, “haNK”). This promotes cell lysis through binding of the Fc region of IgG1 antibodies, either naturally derived in the host or via exogenous delivery of tumor-antigen specific mAbs (i.e., avelumab, trastuzumab, and cetuximab) [80,81]. haNK cells, although simplistic, are impactful, for even if there is no CAR present to target the NK cell to the tumor, they still outperform patients’ endogenous NK cells. In a planned Phase I, 3 + 3 dose escalation study, patients with metastatic or locally advanced solid tumors will receive 2 × 10^9^ haNK cells per infusion (NCT03027128). Primary endpoints are maximum tolerated dose (MTD)/highest tolerated dose (HTD), recording of any dose limiting toxicities (DLTs) and treatment-associated adverse events. Secondary endpoints are objective response rate (ORR), progression-free survival (PFS) by Response Evaluation Criteria in Solid Tumors (RECIST) guidelines and overall survival (OS). This study has completed enrollment but has yet to post results. Other clinical trials using haNK cells, alone or in combination with other immuno-oncology agents, include a Phase II trial in Merkel cell carcinoma (NCT03853317) and a Phase I trial in metastatic or locally advanced solid tumors (NCT03027128).

## 5. Chimeric Antigen Receptor-Expressing NK Cells (CAR-NK)

The use of CAR-T in the clinic was first approved by the FDA in 2018 for children and adults with B cell ALL. However, there are therapeutic difficulties, which include, but are not limited to: (1) manufacturing autologous CAR-T cells is expensive and logistically challenging, (2) CAR-T cells derived from allogeneic donors, even if HLA matched, pose a significant threat of GVHD, (3) efficacy is reduced due to antigen escape, and (4) CAR-T cells are still influenced by the tumor microenvironment (TME) and checkpoint molecules, such as PD-1, TIM-3, LAG-3, CTLA-4 and TIGIT. 

Autologous, allogeneic and NK cell lines can all be engineered to express tumor-targeting CARs (Figure 1C). CAR constructs for NK cells are similar to those designed for T cells. They have an extracellular single-chain fragment variable (scFv) of an antibody directed against a tumor-associated antigen, a hinge region, a transmembrane domain, and intracellular domain(s) generally composed of the signaling subunits of costimulatory molecules, such as CD3ζ and/or CD28. Over the past decades, significant progress has been made in the iterative development of effective CAR constructs. The biology of these generations of CAR constructs is beyond the scope of this review and has been extensively reviewed elsewhere [82]. The cumulation of this work has resulted in several CAR-NK strategies entering the clinic, both in hematologic malignancies and solid tumors. Herein, we will focus on the clinical efficacy of current and prospective CAR-NK cells. 

### 5.1. Hematologic Malignancies

As observed in CAR-T therapies, approaches using CAR-NK treatment have been most successfully applied in hematologic malignancies. As of this writing, there are fewer than 10 clinical trials using CAR-NK cells in the treatment of liquid tumors (Table 1). Below, we will review advances with CAR-NKs in B cell malignancies (CD19-CAR-NK) and AML (CD33-CAR-NK). 

CD19 is an attractive target in B cell malignancies, but CD19 CAR-T therapies have substantial toxic effects and involve complex manufacturing. Liu et al. developed allogeneic cord blood-derived NK cells that have been retrovirally transduced to express an anti-CD19 CAR, endogenous IL-15 and an inducible caspase 9 kill switch. In a Phase I/II dose escalation clinical trial at MD Anderson Cancer Center (NCT03056339), patients with relapsed or refractory CD19 positive cancers were administered 1 × 10^5^, 1 × 10^6^ or 1 × 10^7^ HLA-mismatched anti-CD19-CAR-NK cells. Although the study number was small, 8/11 patients had a clinical response (CR) (73%) and 7/8 responders had complete remission. These responses were not dose dependent, as CR was observed across all doses, with MTD not reached, and CAR-NKs were detected at low levels for at least 12 months. Importantly, there was no observed cytokine release syndrome (CRS) (frequently associated with CAR-T therapy), GVHD or neurotoxicity [83]. Although this study used fresh, HLA-mismatched CAR-NK cells for each patient, the group recently reported the ability to generate >100 doses of CAR-NK from a single cord-blood unit [84], making this a feasible and attractive “off the shelf” therapy if the data can be repeated with cryopreserved CAR-NK cells. 

Additional CAR-NK strategies have been developed targeting CD33, a member of the Siglec family that is expressed on myeloid leukemic blasts in a majority of patients with AML [85]. CAR-T therapy has shown very little success targeting CD33, with observations of severe side effects such as CRS and neurotoxicity [86]. In contrast, the first-in-human clinical trial of CD33-targeting CAR-NK cells reported that they were safe and well tolerated in three patients with relapsed or refractory AML (NCT02944162). This CAR-NK line was derived via lentiviral transfection of NK-92 cells with a CAR targeting CD33, a transmembrane domain and intracellular domain containing two costimulatory molecules, CD28 and 4-1BB (CD33-CAR-NK-92). Although initial concerns were raised about the longevity of irradiated CD33-CAR-NK-92 cells in vivo, the cells were detected one-week post-infusion, and there was a modest, although insignificant, treatment effect. 

Although only select CAR-NK therapies for hematologic malignancies were highlighted in this review, it is important to note there are several other CAR-NKs targeting various TAAs in the preclinical and clinical pipeline, including but not limited to CAR-NKs for use in refractory B cell lymphoma (CD22-CAR-NK, CD19-CAR-NK, CD22/CD19-CAR-NK), diffuse large B cell lymphoma (CD19-t-haNK), T cell ALL (CD7-CAR-NK) [87] and relapsed and refractory multiple myeloma (BCMA-CAR-NK) (NCT03940833) (Table 1).

### 5.2. Solid Tumors

A major obstacle presented to CAR therapy is the lack of efficacy in solid tumors. This is due to poor perfusion to the tumor, TAA heterogeneity and the immunosuppressive tumor microenvironment (TME) that accompanies a solid tumor (Figure 2). CAR-NK therapy attempts to remedy these issues with a limited number of clinical trials employing this technology in the treatment of solid malignancies, several of which will be reviewed here (Table 2). The CAR-NK strategies that have progressed the furthest in solid tumors target the checkpoint molecule programmed cell death protein-1 ligand (PD-L1), as well as common tumor-associated antigens HER2 and mucin1 (MUC1).

PD-L1 is upregulated in several cancer types in the TME and on immune suppressor cells. Infiltrating T lymphocytes often express PD-1, and upon engagement with PD-L1 experience a significant decrease in effector function. Due to the specific expression of PD-L1 on tumor and immunosuppressive cells, PD-L1 was identified as an optimal target for CAR therapies. A novel NK-92 cell line was engineered to express a CAR specific for PD-L1, ER-retained IL-2 and high affinity CD16, termed PD-L1 targeted-haNK (t-haNK) [88]. Exciting preclinical data demonstrated PD-L1 specific efficacy of these cells against 15 tumor cell lines in vitro, and strong anti-tumor efficacy in vivo of triple negative breast cancer, bladder and lung tumors [88]. These data contributed to the QUILT 3.064 Phase I clinical trial evaluating safety and preliminary efficacy of PD-L1 t-haNK in patients with locally advanced or metastatic solid cancers (NCT04050709). Primary outcomes will address MTD/HTD, recommended dose for Phase II, and incidence of DLTs and treatment emergent adverse events. ORR, PFS, and OS will be determined using RECIST and immune-related RECIST as secondary outcomes. This trial is currently recruiting with no results posted. In a planned Phase II study currently recruiting, PD-L1 t-haNK in combination with other agents will be evaluated for safety and efficacy in patients with locally advanced or metastatic pancreatic cancer (NCT04390399). 

HER2 is overexpressed in several cancer types, such as breast cancer [89], gastric cancer [90], esophageal cancer [91,92], ovarian cancer [93] and endometrial cancer [94]. HER2 is also expressed in 80% of glioblastoma tumors and correlates with poor survival [95]. HER2 is an attractive target for CAR therapy because it is not generally present in the healthy adult central-nervous system (CNS) [96]. HER2-CAR-T therapy in glioblastoma has shown trafficking of these cells to the tumor, but the generation of high effector function T cells within the TME results in rapid selection of antigen-loss variants [97,98]. Therefore, a HER2-CAR-NK treatment strategy was developed. In a 3 + 3 dose escalation Phase I clinical trial (CAR2BRAIN; NCT0338978), patients with recurrent HER2-positive glioblastoma, with an already scheduled relapse resection surgery, received 1 × 10^7^, 3 × 10^7^ or 1 × 10^8^ ErbB2-NK-92/5.28z CAR-NK cells/2ml injected intracranially. Due to the blood-brain barrier, CAR-NK cells were delivered through intracranial injection into the resection cavity wall to allow for a high density of effector cells that would otherwise be excluded from the tumor site (Figure 2, left panel). There were no DLTs detected with the first two cell concentrations, with the third (1 × 10^−8^) cell dose currently ongoing. In the subsequent expansion cohort, patients will be eligible to receive up to 12 weekly CAR-NK infusions administered into the resection cavity through a catheter and reservoir implanted during the initial relapse resection surgery. Through use of HER2-CAR-NK cells, and direct injection into the tumor site, ErbB2-NK-92/5.28z CAR-NK will maintain effector function despite the potential antigen loss, which would return cytotoxic function of adoptively transferred cells to that of baseline NK cells. Further studies in glioblastoma are ongoing using EGFR, another well-known cell surface protein overexpressed in 40–60% of glioblastoma tumors [99]. Preclinical studies using NK-92 cells transduced to express a CAR specific for EGFR have shown efficacy in tumor models of breast cancer brain metastasis [100], glioblastoma models and patient derived glioblastoma stem cells [101,102].

MUC1 is a heavily glycosylated transmembrane glycoprotein expressed on ductal epithelial cells, with controversial evidence of its expression on human B and T cells [103,104] (reviewed here: [105]). MUC1 is overexpressed in several cancer types, including adenocarcinoma [106], making it a suitable target for CAR-T therapy. However, CAR-T therapies have proven unsuccessful due to the immunosuppressive TME. To circumvent the immune suppression mediated by the TME, Li et al. engineered NK-92 cells to express a MUC1-CAR with CD28 and CD137 signaling domains and a truncated PD-1 peptide (Muc1-CAR-NK-92). These cells were shown to lyse MUC1 target cells in vitro and in vivo. A Phase I clinical trial (NCT02839954) evaluating the safety and efficacy of Muc1-CAR-NK-92 in patients with MUC1 positive relapsed or refractory solid tumors has been recently completed. Thirteen patients with PD-L1 and MUC1 positive cancers, including but not limited to lung, pancreatic, ovarian and colon cancer, were enrolled in the trial and received 1 × 10^9^ Muc1-CAR-NK-92 cells/infusion. Of the initial 13 patients, three were withdrawn, nine had stable disease and one patient had progressive disease [107]. A primary outcome measure was to determine the toxicity profile of Muc1-CAR-NK-92 cells, and notably, there was no evidence of cytokine storm or bone marrow suppression in any patients on trial. These results support the conclusion that the therapy is safe and mildly effective. 

### 5.3. Early Phase I Trials

In both hematologic malignancies and solid tumors there are several clinical trials that are recruiting, or active but not yet recruiting, that have not yet posted results. Although the trials are immature, they are noteworthy, and it is important to consider the diverse CAR-NK targets and treatment strategies under investigation when discussing the future of CAR-NK therapy in the clinic. Included in this review are CAR-NK targeting NKG2D, prostate-specific membrane antigen (PSMA), and mesothelin (Table 1 and Table 2). 

NKG2D ligands (NKG2DL) are expressed on most human tumor cells, such as ovarian, breast, prostate, colon and bladder cancer as well as leukemias and lymphomas [108]. NKG2DL is also expressed on Tregs, myeloid derived suppressor cells (MDSCs) and epithelium promoting tumor survival and suppression of a robust anti-tumor immune response (reviewed here: [109]). In a Phase I pilot study (NCT03415100), patients with metastatic solid tumors will receive autologous or allogeneic NK cells that are transfected via mRNA electroporation to express a CAR specific for NKG2DL. The primary outcome is number of adverse events with secondary outcome evaluating anti-tumor response to CAR-NK infusions. Some patients enrolled will receive IL-2, delivered subcutaneously, to support in vivo growth of adoptively transferred NK cells. Combination strategies using NKG2DL-CAR-NK infusions with therapies that naturally increase NKG2DL expression on tumor cells through immunogenic modulation, such as localized irradiation [110], should be explored. 

Additional studies are investigating PSMA, a protein that is highly expressed on epithelial cells of the prostate with very low or undetectable levels in epithelium of other tissues and organs. The expression selectivity makes PSMA an ideal candidate for CAR targeting. The safety and efficacy of anti-PSMA-CAR-NK cell infusion in metastatic castration resistant prostate cancer patients is being interrogated in a Phase I, 3 + 3 dose escalation clinical trial (NCT03692663). The occurrence of treatment-related adverse events, defined as Grade 3 or greater, will be the primary endpoint. This trial is not yet recruiting and as such no results have been posted as of this writing. 

Mesothelin is expressed in low levels in mesothelial cells of pericardium, pleura and peritoneum of healthy individuals, but is overexpressed in a variety of cancers, including but not limited to stomach, pancreatic, lung, breast and ovarian cancer [111]. In a single center, single arm, open label clinical trial (NCT03692637), mesothelin-positive patients with stage II-IV epithelial ovarian cancer will receive autologous NK cells engineered to express an anti-mesothelin CAR. The study is designed as a 3 + 3 dose escalation trial where upon enrollment, patient blood is drawn, mononuclear cells are isolated and sent to a manufacturing facility to produce anti-mesothelin CAR-NK cells that will then be infused back into the patient. Primary outcomes will be occurrence of treatment-related adverse events. 

## 6. Next-Next-Generation CAR-NKs

Notable concerns associated with the use of CAR-therapy, T or NK, are the loss of or decrease in expression of targeted TAAs, rendering CAR-T ineffective and returning CAR-NK cytotoxic function to baseline physiologic capacity of native NK cells, and the influence of the microenvironment on CAR-NK (Figure 2). Several research groups are exploring different mechanisms of combating these concerns. Effectively countering antigen loss (Figure 2, middle panel) and the immunosuppressive TME (Figure 2, right panel) would increase the therapeutic lifespan and efficacy of CAR-NK treatment strategies and may be the backbone of next-next-generation CAR-NK therapy. 

### 6.1. Universal CAR (UniCAR)

To avoid antigen escape, Mitwasi and colleagues developed a universal CAR platform (UniCAR) with an on/off switch to improve safety and controllability [112]. This technology was first demonstrated with CAR-T cells specific for E5B9, a peptide epitope from the nuclear antigen La-SS/B. Because this protein is not found on the cell surface, the CAR-T must be directed to the tumor through the use of a bispecific component, called a target module (TM). TMs usually exist as an E5B9 peptide epitope fused to a scFV of an antibody directed against a TAA. TMs that exist on this platform may be combined with additional TMs to target multiple TAAs simultaneously, inducing heterogeneity in the peptides the CARs are responding to without the risk of off-target effects (Figure 2, middle panel). Mitwasi et al. [112] generated a UniCAR-expressing NK-92 cell line and a TM where the E5B9 epitope is connected to an anti-GD2 mAb on an IgG4 backbone. Although the IgG4 isotype significantly increases the half-life and persistence in vivo, the concern about an on-target/off-tumor effect remains low since NK-92 cells are irradiated prior to infusion. The IgG4 backbone was also chosen because it weakly activates complement C1q, reducing the ability to trigger ADCC in comparison to IgG1 and IgG3. GD2 is highly overexpressed in a variety of human tumors, and one of the only immunotherapy targets in neuroblastoma [113,114]. In fact, a mAb against GD2, dinutuximab, was approved for patients with high risk neuroblastoma [115]. UniCAR NK-92 targeted to GD2-expressing neuroblastoma and melanoma cells can lyse and secrete IFNγ in vitro, as well as antigen-specific lysis of Panc-89 cells directed against EGFR expressed on the cell surface. These data demonstrate proof of concept for further preclinical and clinical experimentation. 

### 6.2. Shifting Signals

Another challenge facing CAR therapy is the TME (Figure 2, right panel). It is as of yet unknown whether CAR-NK cells can effectively traffic to a solid tumor, and whether they are as susceptible to the immunosuppressive TME as CAR-T cells. To avoid the immunosuppressive TME, Wang et al. constructed NK-92 cells expressing a modified CAR consisting of extracellular TGFβRII fused to the intracellular signaling domain of the activating receptor, NKG2D (TGFβRII-NKG2D, termed NK-92-TN) [116]. This fusion CAR, in essence, switches the signal delivered from TGFβ generated from the immunosuppressive TME into an activating signal. In vitro, NK-92-TN cells are resistant to TGFβ suppression, exhibiting no NKG2D downregulation and display increased killing capacity and IFNγ production following TGFβ co-culture. Further interrogation demonstrated that treatment of NK-92-TN with TGFβ did not result in canonical TGFβR signaling, as evidenced by the absence of pSMAD2. In a transwell assay, NK-92-TN cells demonstrate increased migration to tumor cells expressing TGFβ, and have increased expression of chemokine receptors (CCR3, CCR6, CXCR4 and CX3CR1) in comparison to vector control NK-92 cells. In fact, the presence of TGFβ led to a further enhancement of NK-92-TN cytolytic properties, chemoattraction to tumor cells and prevention of CD4 differentiation to Tregs in a hepatocellular carcinoma xenograft model. Although the in vitro data look promising, in vivo, NK-92-TN anti-tumor activity was moderate at best, with a minimal reduction of tumor weight at endpoint, and a slight, albeit significant increase in tumor infiltrating lymphocytes in animals treated with NK-92-TN. These results could be due to the model being used (athymic nude Balb/c mice inoculated with SMMC7721) or that only one dose was administered. Use of humanized mouse models and repeated injections may more closely mirror what is hoped to be observed in the clinic and include the effects these cells could have on the immune populations surrounding them. 

## 7. Conclusions

Harnessing cells of the immune system to fight various malignancies has been widely explored, with the employment of immunotherapy revolutionizing cancer treatment. However, only a fraction of patients achieve durable clinical responses. Adoptive cell therapy with T cells has undergone extensive research and demonstrated clinical efficacy, and a growing body of evidence suggests that NK cells are also safe and efficacious. While decades of research have demonstrated the difficulties surrounding adoptive cell transfer of allogeneic and autologous NK cells, the data have also found that NK cells represent a pool of innate immune cells poised for rapid tumor clearance. The development of novel, immortalized NK cell lines for off-the-shelf therapy, as well as new gene-editing techniques to arm NK cells with additional mechanisms for targeting tumor cells (CAR) and increasing their cytotoxic potential (endogenous cytokine production) are pushing the field forward in exciting and important ways. It is clear that NK cell transfer is a powerful weapon in the fight against cancer, and that NK cells will play a significant role in the future of immuno-oncology clinical strategies. 

## Figures and Tables

**Figure 1 cancers-12-02131-f001:**
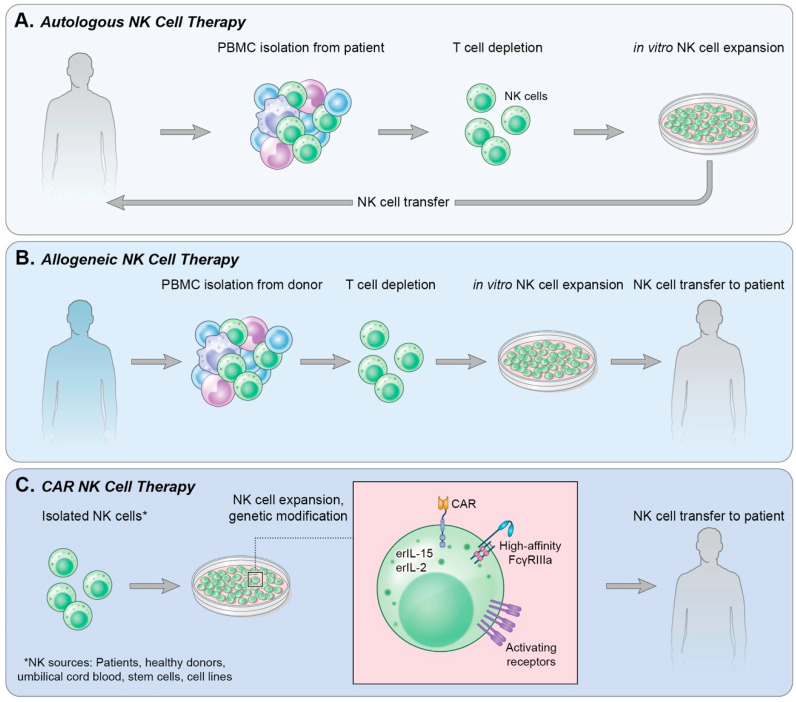
Methods of NK cell therapy. (**A**) Autologous NK cell therapy uses a patient’s own endogenous NK cells isolated from peripheral blood. T cells are depleted from the culture followed by in vitro expansion of NK cells and reinfusion into the patient. (**B**) Allogeneic NK cell therapy isolates NK cells from a healthy donor’s PBMCs in a process similar to that in (**A**); however, following NK cell expansion in vitro, the cells are infused into a patient. (**C**) CAR NK therapy can use NK cells derived from various sources. NK cells are then genetically modified to express receptors and cytokines of choice and expanded. The CAR NK cells are then infused into a patient.

**Figure 2 cancers-12-02131-f002:**
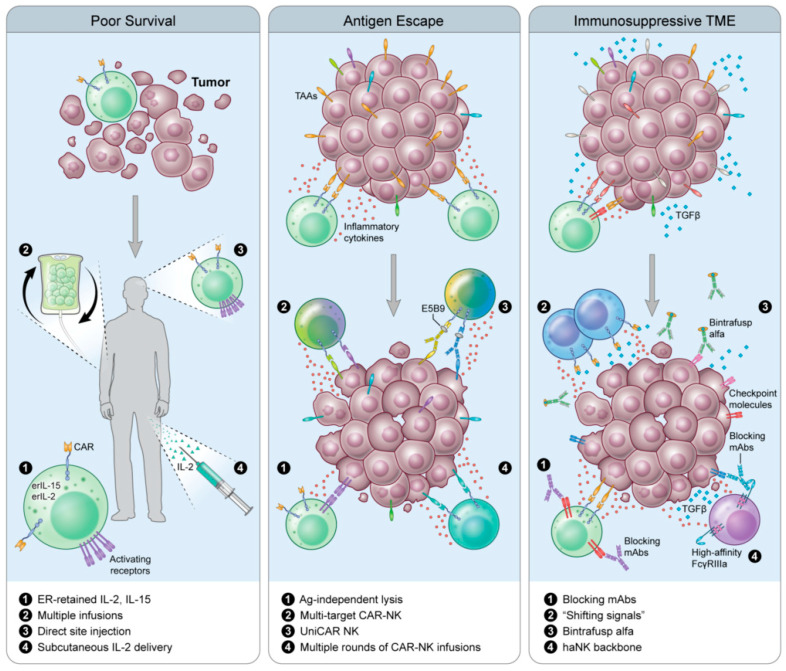
Overcoming CAR-T resistance with genetically modified NK cells. (**A**) Poor survival of adoptively transferred cells can be overcome in a variety of ways such as ER-retained cytokines, multiple infusions of CAR NKs, direct site injection and subcutaneous delivery of IL-2 to promote in vivo survival. (**B**) Antigen escape can be overcome by NK cells’ ability to lyse target cells in an antigen-independent manner, generation of multi-target CAR-NKs, universal CAR-NKs directed to multiple tumor antigens simultaneously or in succession, and the ability to infuse multiple rounds of CAR-NK cells. (**C**) The immunosuppressive TME is a challenge for CAR-T and CAR-NK cell therapy. Some strategies to overcome this include combining CAR-NK with blocking mAbs, using CAR-NKs with receptors specific for inhibitory cytokines that signal through an activating receptor domain, sequestering of inhibitory cytokines through combinatorial use of bintrafusp alfa (a bifunctional protein targeting TGFβ and PD-L1), and having CAR-NK cells that express the high affinity FcγRIIIa receptor to increase ADCC potential.

**Table 1 cancers-12-02131-t001:** Clinical trials in Hematologic Malignancies using CAR-NK therapy.

Clinical Trial	Indication	Status	NK	NK Source	Trial Phase	*n*	Outcomes/Key Points
NCT03056339	B Lymphoid Malignancies	Recruiting	iC9/CAR.19/IL-15 NK	Umbilical Cord Blood	Phase I/II	36	Safety, relative efficacy; ORS (CR/PR), persistence of infused NK, comprehensive immune reconstitution studies
NCT02944162	Relapsed and Refractory CD33+ Acute Myeloid Leukemia	Unknown	anti-CD33CAR NK	NK-92	Phase I/II	10	Safety; anti-leukemia response; determine anti-NK response
NCT03692767	Relapsed and Refractory B Cell Lymphoma	Not Yet Recruiting	anti-CD22CAR NK	Undisclosed	Early Phase I	9	Occurrence of TRAE per CTCAE v4.0
NCT03690310	Relapsed and Refractory B Cell Lymphoma	Not Yet Recruiting	anti-CD19CAR NK	Undisclosed	Early Phase I	9	Occurrence of TRAE per CTCAE v4.0
NCT03824964	Relapsed and Refractory B Cell Lymphoma	Not Yet Recruiting	anti-CD19/CD22 CAR NK	Undisclosed	Early Phase I	10	Occurrence of TRAE per CTCAE v4.0
NCT02892695	CD19+ Leukemia and Lymphoma	Unknown	anti-CD19CAR NK	NK-92	Phase I/II	10	TRAEs; Objective Response Rate
NCT04052061	Diffuse Large B Cell Lymphoma	Not Yet Recruiting	anti-CD19t-haNK	haNK	Phase I	18	MTD, HTD, RP2D, incidence of DLTs and TRAE per CTCAE v5.0; ORR, PFS, OS
NCT02742727	CD7+ Leukemia and Lymphoma	Unknown	anti-CD7CAR-pNK	NK-92	Phase I/II	10	Occurrence of TRAE per CTCAE v4.0; CR and persistence of NK
NCT03940833	Relapsed and Refractory Multiple Myeloma	Recruiting	anti-BCMACAR NK-92	NK-92	Phase I/II	20	Occurrence of TRAE per CTCAE v4.0
NCT03579927	B Cell Non-Hodgkin Lymphoma	Withdrawn	CAR.CD19-CD28-zeta2A-iCasp9-IL15	Umbilical Cord Blood	Phase I/II	0	Safety, relative efficacy; RFS, OS, persistence of infused NKs

ORS: Objective Response Rate. CR: Complete Response. PR: Partial Response. TRAE: Treatment Related Adverse Event. CTCAE: Common Terminology Criteria for Adverse Event. MTD: Maximum Tolerated Dose. HTD: Highest Tolerated Dose. RP2D: Recommended Phase 2 Dose. DLTs: Dose-Limiting Toxicities. ORR: Objective Response Rate. PFS: Progression-Free Survival. OS: Overall Survival. RFS: Relapse-Free Survival. DCS: Disease Control Survival. RECIST: Response Evaluation Criteria in Solid Tumors. QoL: Quality of Life. MFD: Maximum Feasible Dose.

**Table 2 cancers-12-02131-t002:** Clinical trials in Solid Tumors using CAR-NK therapy.

Clinical Trial	Indication	Status	NK	NK Source	Trial Phase	*n*	Outcomes/Key Points
NCT04050709	Locally Advanced or Metastatic Solid Cancers	Recruiting	anti-PD-L1t-haNK	haNK	Phase I	16	MTD, HTD, RP2D, incidence of DLTs and TRAE per CTCAE v5.0; ORR, PFS, OS
NCT04390399	Locally Advanced or Metastatic Pancreatic Cancer	Not Yet Recruiting	anti-PD-L1t-haNK	hanK	Phase II	268	PFS per RECIST V1.1; ORR, CR, DCR per RECIST, OS, QoL by Patient Reported Outcomes
NCT03383978	HER2+ Glioblastoma	Recruiting	NK-92/5.28.z	NK-92	Phase I	30	Occurrence of TRAE per CTCAE v4.03, MTD/MFD, persistence of NK, cytokine profile in CSF; ORR, PFR, OS and anti-NK response
NCT02839954	MUC1+ Relapsed or Refractory Solid Tumors	Unknown	anti-MUC1CAR-pNK	PlacentalHSC-derived	Phase I/II	10	Occurrence of TRAE per CTCAE v4.0; ORR (PR/CR)
NCT03415100	Metastatic Solid Tumors	Recruiting	anti-NKG2DL CAR NK	Autologous or Allogeneic	Phase I	30	Number of adverse events; anti-tumor response
NCT03692663	Castration-Resistant Prostate Cancer	Not Yet Recruiting	anti-PSMACAR NK	Undisclosed	Early Phase I	9	Occurrence of TRAE per CTCAE v4.0
NCT03692637	Epithelial Ovarian Cancer	Not Yet Recruiting	anti-Mesothelin CAR NK	Autologous	Early Phase I	30	Occurrence of TRAE per CTCAE v4.0
NCT03940820	Solid Tumors	Recruiting	anti-ROBO1CAR NK	Undisclosed	Phase I/II	20	Occurrence of TRAE per CTCAE v4.0
NCT03941457	Pancreatic Cancer	Recruiting	anti-ROBO1 BiCAR NK	Undisclosed	Phase I/II	9	Occurrence of TRAE per CTCAE v4.03
NCT03931720	Malignant Tumors	Recruiting	anti-ROBO1 BiCAR NK/T	Undisclosed	Phase I/II	20	Occurrence of TRAE per CTCAE v4.0

ORS: Objective Response Rate. CR: Complete Response. PR: Partial Response. TRAE: Treatment Related Adverse Event. CTCAE: Common Terminology Criteria for Adverse Event. MTD: Maximum Tolerated Dose. HTD: Highest Tolerated Dose. RP2D: Recommended Phase 2 Dose. DLTs: Dose-Limiting Toxicities. ORR: Objective Response Rate. PFS: Progression-Free Survival. OS: Overall Survival. RFS: Relapse-Free Survival. DCS: Disease Control Survival. RECIST: Response Evaluation Criteria in Solid Tumors. QoL: Quality of Life. MFD: Maximum Feasible Dose.

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
