# Peer review of "Natural Born Killers: NK Cells in Cancer Therapy"

_cancers, 2020, doi:10.3390/cancers12082131_

Round 1

Reviewer 1 Report

Optional: An updated on the Wels study with Her-2 new NK-92 cells is mentioned in: https://www.frontiersin.org/articles/10.3389/fimmu.2019.02683/full

Also suggest to quote an ASH abstract from 2018 describing a t-haNK for CD19 engineered to express the CCR7 homing receptor (abstract #116890)

Author Response

July 20, 2020

Re: cancers-875047

Natural Born Killers: NK Cells in Cancer Therapy

Specific Points:

NOTE: Changes in BLUE

REVIEWER 1:

  1. An updated on the Wels study with Her-2 new NK-92 cells is mentioned in: https://www.frontiersin.org/articles/10.3389/fimmu.2019.02683/full

Response:

            We thank the reviewer for this suggestion. The article mentioned is a review (CAR-Engineered NK Cells for the Treatment of Glioblastoma: Turning Innate Effectors Into Precision Tools for Cancer Immunotherapy). We have now referenced the trials discussed in the review (manuscript references 99-102).

  1. Also suggest to quote an ASH abstract from 2018 describing a t-haNK for CD19 engineered to express the CCR7 homing receptor (abstract #116890) all in-vitro

Response:

            We understand. The noytes ASH abstract describes the in-vitro utilization of  a t-haNK for CD19 engineered to express the CCR7 homing receptor. To focus our review, we limited our comments to clinical data at the expense of in-vitro only data. If the reviewer insists, we can add it.

Reviewer 2 Report

Well written review article on NK cells in cancer therapy that includes several pertinent recent studies. Concise and very easy to read. Figures are well-done, clear and self-explanatory.

Some minor comments that should be addressed:

  1. please include trade names of the drugs in lines 39 and 40.
  2. include the word "cell" after NK in lines 83, 86 and 88.
  3. Should include a brief discussion on the cytotoxic molecules referred to in line 128.
  4. the authors refer to several CAR-NK therapy in lines 324-329. Are these using NK-92 cell line or primary human NK cels. A sentence should be added to clarify this.
  5. what does t-ha-NK refer to in line 356 ? please clarify.
  6. spell out TME in line 332.
  7. please include appropriate reference for the statement in lines 465-467.

Author Response

July 20, 2020

Re: cancers-875047

Natural Born Killers: NK Cells in Cancer Therapy

Specific Points:

NOTE: Changes in BLUE

REVIEWER 2: 

  1. Please include trade names of the drugs in lines 39 and 40.

Response:

            We appreciate this comment. However, as United States Federal Employees, we are forbidden form listing trade names, as this may imply an endorsement of a particulat drug by the government. We are strongly encouraged to use the generic names for drugs.

  1. include the word "cell" after NK in lines 83, 86 and 88.

Response:

            As suggested, this has now been added.

  1. Should include a brief discussion on the cytotoxic molecules referred to in line 128.

Response:

            Thank you for this comment. As suggested, this has now been added.

  1. the authors refer to several CAR-NK therapy in lines 324-329. Are these using NK-92 cell line or primary human NK cells. A sentence should be added to clarify this.

Response:

            As recommended, we have now clarified this in Table 1.

  1. what does t-ha-NK refer to in line 356 ? please clarify.

Response:

            Thanks. This has now been clarified.

  1. spell out TME in line 332.

Response:

            Thank you. As suggested, this has now been clarified.  

  1. please include appropriate reference for the statement in lines 465-467.

Response:

This has now been completed.